# Metal Microelectromechanical Resonator Exhibiting Fast Human Activity Detection

**DOI:** 10.3390/s23218945

**Published:** 2023-11-03

**Authors:** Francesc Torres, Arantxa Uranga, Núria Barniol

**Affiliations:** Electronic Engineering Department, Universitat Autònoma de Barcelona, Edifici Q, Campus UAB, 08193 Cerdanyola del Valles, Spain; arantxa.uranga@uab.cat (A.U.); nuria.barniol@uab.cat (N.B.)

**Keywords:** CMOS, humidity sensor, low power, MEMS, monolithic integration, resonators, non-contact detection

## Abstract

This work presents a MEMS resonator used as an ultra-high resolution water vapor sensor (humidity sensing) to detect human activity through finger movement as a demonstrator example. This microelectromechanical resonator is designed as a clamped-clamped beam fabricated using the top metal layer of a commercial CMOS technology (0.35 μm CMOS-AMS) and monolithically integrated with conditioning and readout circuitry. Sensing is performed through the resonance frequency change due to the addition of water onto the clamped-clamped beam coming from the moisture created by the evaporation of water in the human body. The sensitivity and high-speed response to the addition of water onto the metal bridge, as well as the quick dewetting of the surface, make it suitable for low-power human activity sensing.

## 1. Introduction

Human activity sensors have great importance for many applications, from movement and proximity sensing to breath monitoring; they are capable of offering valuable platforms for human–machine non-contact interfaces and valuable insights into health conditions (i.e., respiratory rate and sleep depth to detect sleep apnea, or finger movement tapping rates for early Parkinson disease detection [1]). In parallel, high-performance humidity detection has taken an important development. Particularly, real-time humidity detection allows us to design sensors for human activity such as breath monitoring or non-contact finger movement [2,3,4,5,6,7].

In both cases, the level of humidity is changed locally due to the water vapor contents in any human exhalation during breathing, and the sharp gradient of humidity around the fingertip [2] due to skin transpiration.

Humidity sensors can provide a convenient and effective means of human activity monitoring, with a non-intrusive, non-contact nature, and high sensitivity, making them a popular choice.

To detect humidity, there are many proposals in the literature, from capacitive [1] or resistive sensors [8,9], surface acoustic waves (SAW) [10,11,12], or sensors based on mechanical resonators [13,14,15,16,17,18,19,20], among others. Most of them require a specifically designed geometry, and some kind of functionalization using a sensitive layer or special materials (2D materials [7], nanoforest structures [12], etc.) to increase the final achievable resolution.

As an example of resistive sensors, the authors of [9] present an interesting and easy-to-fabricate flexible sensor based on cellulose with simple pencil-on-paper hand-drawn interdigitated electrodes. The paper is treated with a solution of NaCl, and a response time of near 29 s is obtained when the sensor is cycled from 15% to 40% of relative humidity (RH). This sensor is used for finger movement detection and respiratory monitoring, among other uses.

The authors of [10] present a very interesting sensor based on a SAW resonator with graphene oxide and MoS_2_ (as a sensing layer) films. Graphene oxide is used to prevent the short circuiting of the electrodes by a high conductive MoS_2_ film in high relative ambient humidity. The authors have obtained a high sensitivity of 12.41 kHz/%RH that corresponds to 0.005% of the change of frequency of resonance per %RH.

One of the main uses of MEMS resonators (cantilevers, bridges, or plates) is as a gravimetric sensor that detects mass changes through the change of their resonant frequency. This frequency change can be sensed mainly using well-known optical, piezoelectrical, or capacitive transductions, among others. The capacitive transduction is widely used and is a perfect candidate to be monolithically integrated with a signal processing and readout circuit made with CMOS commercial technology.

For example, in [17] the authors present a very interesting 4-anchored plate oscillator integrated monolithically with CMOS technology. The plate has dimensions of 41 μm × 10 μm with distributed holes to facilitate the releasing. The authors have obtained a sensitivity of 0.8% in frequency of resonance decreasing per %RH.

In [18] the authors have proposed a piezoelectric cantilever (300 × 1000 μm^2^) with an MoS_2_ film deposited onto it, with a fast response time of 1 s, and 2.8 s of recuperation time and with a good sensitivity of 0.3% in RH. This cantilever is implemented with machine learning algorithms to perform the analysis of the results. 

The authors from [19] present a commercial double cantilever system, one that is 110 μm in length, and the other that is 50 μm in length, which is used as a reference. The authors use piezoresistive detection and obtain a mass sensitivity of near 300 fg/Hz. The sensor operates in a dynamical mode using a piezoelectric material to shake the cantilevers. The detection is performed by measuring both the frequency shift due to the water (mass) addition onto the cantilever, and the increase of the amplitude of vibration due to the water addition.

In [20], a CMOS-MEMS humidity sensor is presented, based on a moving plate of 400 μm × 400 μm of surface, and using a CMOS technology TiO_2_ layer to absorb the water. The changes in the amplitude of vibration (detected using piezoresistive effect) are used for sensing. Also, the sensor is actuated using electrothermal transduction and is operated in dynamical mode. The authors have obtained a sensitivity of 0.1 mV/%RH.

The metal sub-micrometric (clamped-clamped) cantilever resonator, which is integrated with CMOS technology, has been demonstrated to be a very sensitive mass sensor [14] and humidity sensor [15], acting as a mass sensor (water adsorption at the resonator surface). A comparison of this system with other systems based on optical and piezoelectric transduction was performed.

Water absorption onto the clamped-clamped beam produces a change of the mass of the resonator, and consequently, a change of the resonance frequency occurs, following (1):(1)$$\Delta f=-\frac{f_{0}}{2m_{eff}}\Delta m,$$
where *f*_0_ is the resonance frequency of the clamped-clamped beam, Δ*m* is the added mass, and *m_eff_* is the effective mass of the clamped-clamped beam (0.4*m*_0_ where *m*_0_ is the mass of the bridge) [21].

Due to its low effective mass and high frequency of resonance, the sensor is capable of detecting humidity with a high sensitivity without the need of any functionalization layer. Due to this, we propose this MEMS resonator as a highly sensitive, non-contact and fast human activity detector.

## 2. Sensor Design and Fabrication

The MEMS sensor utilized in this study is 18 μm long, 600 nm wide, and an 850 nm thick metal bridge resonator. It was fabricated using the top metal layer of a commercial 0.35 μm CMOS technology provided by Austria Microsystems (AMS) (Premstaetten, Austria) (3.3 V, 4-metal 2-poly), as shown in Figure 1. For the top metal we have used the minimum width possible offered by this technology (0.6 μm). The design of the system does not incorporate a passive layer, enabling a straightforward post-CMOS process without the need of any mask. This process involves the wet etching of the intermetal oxide (sacrificial layer) to release the bridge, immersing the chip into hydrofluoric acid for a few seconds, and then rinsing it with deionized water and isopropyl alcohol to avoid possible stiction of the beam into electrodes or bottom layers. Since the resonator is located on the top metal layer, this release process is expedited, allowing for the maximum preservation of the metal by minimizing its etching rate compared to that of the intermetal oxide. The resonator is capacitively connected to a CMOS readout amplification circuitry through the sensing electrode (transimpedance amplification with a gain of 1 MΩ at 10 MHz [22] and references therein), facilitating an easy detection of the bridge’s frequency response.

Sensing is performed through the motional current measurement and then amplified (transimpedance amplification), generated between the bridge and read-out electrode due to the bridge movement, which can be approximated using (2): (2)$$I≅V_{dc}\frac{\partial C}{\partial t}+C_{0}\frac{\partial V_{ac}}{\partial t},$$

This assumes that the excitation voltage *V_ac_* is much smaller than the driving voltage *V_dc_* applied to the bridge (capacitive transduction). In (2), the first term corresponds to the motional current, and the second term is the parasitic current; *∂C*/*∂t* is the bridge-readout electrode variable capacitance, and *C*_0_ is the static capacitance between the excitation and readout electrodes.

Motional current measurements were performed using a two-port configuration, i.e., by applying *V_dc_* to the metal beam and *V_ac_* to the excitation electrode and the CMOS amplifier connected to the reading electrode. This configuration minimizes the parasitic current, minimizing the signal level measured out of the resonance, thus enhancing the resonance detection (getting a better resonance peak in contrast with the signal out of resonance), and secondarily, moving the antiresonance peak of the curve (minimum magnitude value peaks shown in Figure 2) away from the resonance (maximum magnitude values shown in Figure 2) curve, allowing for a clear measurement of the frequency of resonance.

This specific resonator was previously evaluated as a humidity sensor in [15], demonstrating excellent performance with a high degree of linearity within the range of 15% to 85% relative humidity (RH). Additionally, it exhibited a very high sensitivity of 3.5 kHz per %RH. Sensor linearity is crucial to accurately measure relative humidity, providing a straightforward means of quantification. 

The sensor limit detection is influenced not only by its sensitivity but also by the transduction and readout circuit employed. By itself, the sensitivity of the system relies on the water adsorption capability of the metal used, which is relatively limited. While this characteristic alone may seem unfavorable, it does not represent a significant obstacle to achieving a high-quality sensor due to the sub-micrometric dimensions of the bridge. The resonant frequency of the resonator is highly responsive to minute mass changes, surpassing the moderate water vapor adsorption capability of the metal layer. The mass sensitivity of the sub-micrometric clamped-clamped beam is 4 ag/Hz (around 1.4·10^3^ molecules/Hz). This exceptional sensitivity eliminates the need for an additional sensing layer, thereby simplifying the final device. 

This CMOS integrated bridge resonator is presented here as a good candidate for human activity sensing through the variation in ambient humidity, which is a consequence of water evaporation produced by finger proximity.

## 3. Results

In this study, we employed an open-loop configuration with a two-port connection to conduct our measurements. The MEMS resonator was connected to a DC voltage source (Vdc, Keithley 2200-72-1 power supply, Solon, OH, USA), while the actuation electrode was driven by the AC signal (Agilent network analyzer E5100A, Santa Clara, CA, USA). The read-out electrode was directly connected to a dedicated CMOS low noise amplifier, integrated closely with the resonator (not seen in the SEM image of Figure 1). This configuration effectively reduces the parasitic capacitance, thereby improving the detectability of the system, as demonstrated in [23]. To perform the measurements, we utilized a network analyzer (Agilent E5100A), which facilitated accurate data acquisition. Except for the Figure 2 measurements, the sensing measurements along the paper were performed using a Vdc = 30 V and an AC power of −30 dBm (approximately 1 V_RMS_). All measurements were performed at ambient conditions (27 °C and 28% RH) and no humidity chamber was used.

To begin, the frequency response of the MEMS bridge resonator was thoroughly characterized in terms of the magnitude and phase at various bias voltages, as illustrated in Figure 2. The resonant frequency of the bridge was determined to be around 12.8 MHz for Vdc bias of 30 V, used in the measurements along the paper. 

In this work, human activity detection is achieved by monitoring the water evaporation from the skin of the human body. When the sensor is placed in proximity to the human body or a specific body part, it can detect changes in the surrounding humidity, effectively serving as an activity detector (see the sketch in Figure 1, Right). The addition of mass to the bridge, resulting from water adsorption, induces a shift in the resonant frequency towards lower frequencies (see Equation (1)), offering various opportunities for human activity detection. Specifically, we measured the phase shift and magnitude shift in the frequency response of the MEMS resonator to detect finger movements. 

Schematically, the measurement procedure follows the steps listed below:(a)First, we have selected a specific point on the magnitude or phase curves corresponding to the resonant frequency of the bare bridge (without added water, and with the sensor far from human body). For the magnitude curve, the point is at the maximum value of the curve. For the phase curve, this point is at the maximum slope of the phase curve (see Figure 2).(b)After that, the measurements are conducted with a frequency sweep span of zero. By fixing the frequency and recording the phase or magnitude values at the chosen point, a straight (horizontal) line can be obtained on the magnitude/phase versus the measurement event plot (see Figure 3, baseline, as an example). This line will maintain a constant measured value until a mass addition event occurs.(c)When we approximate a human body part, in our case the fingertip, or breathing over the sensor, the frequency response curve shifts to the left (towards lower frequencies) due to the added mass (water) and therefore, the measurement (with a frequency sweep span of zero) reflects this change by altering the constant value (magnitude or phase) to a lower value (see Figure 3, wet foam, as an example). When the fingertip is moved far away or we stop breathing onto the bridge, the measurement returns to the initial value, showing a dewetting process of the metal bridge.

To obtain the plots of the different figures we have corrected the values obtained using the baseline signal level in order to show only the relative shifts (magnitude or phase) with respect to the baseline. It is important to point out that measurements can be performed using the phase or the magnitude curves; there is no special reason to choose a magnitude or phase measurement, except, as we will see in the following sections, if high sensitivity is required, in which case it is better to choose a phase instead of magnitude measurement.

### 3.1. Foam Measurements

To ascertain whether the response of the microbridge sensor is indeed attributed to humidity (water addition to the bridge) or another phenomenon (for example, a temperature change), we conducted preliminary measurements. Initially, we selected the maximum point on the magnitude curve and fixed the frequency to perform a zero-span measurement. To simulate different scenarios, we employed a foam material and repetitively moved it closer to and further away from the sensor (using a manual stage). Using foam material is advantageous because, in addition to accumulating water in its porous structure, it does not generate heat as the human body does, which enables us to separate the effects of humidity on the metal bridge from the thermal effects. 

In Figure 3, two distinct measurements using foam are presented. The first measurement involved dry foam, where the magnitude value exhibited only slight changes compared to the baseline. Conversely, the second measurement utilized wet foam, resulting in a significant alteration of the magnitude value when the foam was brought closer to the sensor (from a few centimeters down to a few millimeters from the sensor). These findings indicate that the changes in magnitude are indeed correlated with the presence of water, confirming the sensitivity of the sensor to humidity variations.

### 3.2. Sensor Calibration (Phase Shift versus Distance)

Once we have stablished that the phase (or magnitude) shift measurement is useful to detect humidity changes, we proceeded to calibrate the response of the microbridge based on the distance from a water source. In order to accomplish this, we conducted measurements by fixing the frequency at the resonance value of the resonator and, measuring the phase shift, varying the distance between our system and a water-filled tank with a manual micropositioning system. The measurements were taken at distances ranging from 10 mm to 3.5 mm (see the schematic in Figure 4a). To obtain each phase shift point, we computed the average value over time (14 s), which is shown in the Figure 4 inset graphs; the statistical error is presented as error bars.

To facilitate the measurements of our CMOS systems, our chip was wire-bonded on a home-made printed circuit board with SMA connectors. The bondings employed use non-electrically protected aluminum connections. Due to the limitations of this setup, we had to limit our measurements to a minimum distance of 3.5 mm over the water-filled tank to ensure the integrity of the system and avoid the contact of the water and wire-bondings. The results of these measurements are depicted in Figure 4.

From the observed data, it is evident that as the distance decreases from 6 mm to smaller distances, there is a significant change in phase. As the system moves closer to the water tank, the phase value loses stability, exhibiting values, in the time domain, that smoothly vary in a non-regular form instead of in a straight line (see Figure 4 insets). This behaviour can be attributed to the dynamic equilibrium of water absorption and desorption onto the bridge, resulting in fluctuations in the phase measurement.

It is worth noting that the top metal used in the bridge construction does not possess specific water-absorption properties. The presence of a dynamic water sheet over the bridge leads to a continuous formation and evaporation of the water layer. This aspect may not be critical for human activity detection, such as the finger movement detection proposed in this work.

### 3.3. Application: Finger Movement

In this section, we discuss the experiments that were performed to prove the ability of our system to detect finger movements. We use both the phase and the magnitude shift as measurement variables.

#### 3.3.1. Phase Shift Detection

Figure 5 illustrates the detection of finger movement using phase as a measurement parameter. In these measurements, the frequency was fixed at the resonance point, specifically at the maximum slope on the phase curve. During the experiment, the finger was repeatedly moved closer to and further away from the microbridge sensor within a few centimeters. Additionally, while maintaining a constant distance in the z-direction (perpendicular to the sensor plane), the finger was moved left and right (5 cm of lateral travel). The minimum distance between the finger and sensor was fixed to few millimeters (taking into account a variability of the distance due to non-precise finger movements) to preserve chip-to-board bondings. Remarkably, both types of movements resulted in a similar response from the bridge, allowing us to state that the resonator has an omnidirectional response that is only dependent on the radial distance to the bridge.

As the humidity level increases, the MEMS resonator experiences a reduction in its resonance frequency, resulting in a decrease in phase (negative phase shift) when it is measured at the dry device resonant frequency. To detect the resonant frequency variations, and therefore the %RH, we can utilize the slope of the frequency response curve. In Figure 2, at a DC voltage of 30 V, the slope of the phase curve (at the resonant point) is approximately −5.7·10^−4^ deg/Hz.

When the finger is moved closer to the detector, reaching a distance of a few millimeters above it, the phase experiences a decrease of approximately 8 degrees (in this example). This decrease corresponds to an approximate frequency shift of 14 kHz, a variation of 4 in %RH, indicating a mass addition of around 56 femtograms (equivalent to approximately 1.7·10^9^ molecules of water), based on the calibration conducted in [15] (3.4·10^−11^ g·cm^−2^·Hz^−1^). Proportionally to the resonant frequency, Δf/f_0_, this shift amounts to around 1100 ppm.

Conversely, when the finger is moved further away, the humidity decreases, and the phase value rapidly recovers. This phenomenon demonstrates the efficient water desorption capability of the metal bridge, which corresponds to its low humidity hysteresis cycle. 

The phase shift slope observed in Figure 5 is approximately 22.2 deg/s, which translates to 39 kHz/s. This remarkable slope enables the sensor to achieve a high sensitivity and rapid response.

In addition to the finger movement measurements described earlier, we conducted further experiments using a commercial gyroscope (LSM6DSL from STmicroelectronics, Geneva, Switzerland) with a resolution of 6.11·10^−4^ rad/s and a gravity sensor (Samsung Gravity Sensor, Suwon, Republic of Korea) with a resolution of 0.001 g. We utilized the gyroscope’s capabilities to measure angular movements, specifically the angular velocity that results from moving the fingertip up and down (bending the finger). We also measured the g-force-related parameters. For these measurements, we selected a fixed point on the phase curve corresponding to the resonance frequency of the bridge.

Figure 6 and Figure 7 illustrate comparisons that were observed between the micrometric bridge response and the measurements obtained from the commercial gyroscope. There is a strong correlation between the phase changes in the bridge measurement and the corresponding gyroscope responses.

Figure 6 specifically compares the bridge response with the angular velocity of the fingertip. It is evident that the maximum phase shift points in the microbridge response coincides with a point situated between the consecutive minimum and maximum angular velocity values. When the fingertip approaches the sensor, the angular velocity reaches a maximum negative value. As the fingertip movement is halted near the sensor bridge, the angular velocity value becomes zero. At this point, the microbridge response presents the minimum phase measurement value (maximum phase shift), corresponding with the point where the finger is closer by. Subsequently, as the fingertip is retracted, the angular velocity reaches a maximum positive value, and the phase measurement returns to its original value.

This observation highlights the close relationship between the phase measurements of the microbridge and the angular velocity of the fingertip as captured by the gyroscope. 

Another notable observation from Figure 6 is that the maximum and minimum values of the angular velocity tend to remain relatively consistent. However, this is not the case for the phase measurements obtained from the bridge response. The bridge response is highly dependent on the proximity of the fingertip, and its sensitivity increases exponentially with decreasing distance. As a result, even when the fingertip is moved at a maximum angular velocity, deep peaks in the bridge measurements cannot be achieved unless the finger is brought in close proximity to the microbridge, within a few millimeters.

This fact highlights the crucial role of distance in influencing the sensitivity and magnitude of the bridge response. Significant changes in the bridge measurements can be obtained only when the finger is brought close to the microbridge, where the proximity effect becomes more pronounced.

Figure 7 presents the acceleration (g-force) measurement, which complements the angular velocity measurement discussed earlier. In this case, we focused on the y-direction g-force component, as specified by the commercial sensor’s capabilities. In the y-axis direction, which represents one of the in-plane directions perpendicular to the Earth’s gravity, the g-force value is close to zero when the fingertip is in a neutral position. However, as the finger is tilted, the g-force value deviates from zero and reaches a maximum negative value at the maximum tilt angle.

To provide some context, a g-force value of 0.5 corresponds to a fingertip tilt angle of approximately 45 degrees. As the fingertip is retracted and returns to a position perpendicular to the Earth’s gravity, the g-force value returns to zero. Just before initiating the finger retraction, when the fingertip is closest to the microbridge, we observe the minimum value of the phase measurement at our sensor. Consequently, as the g-force value for the y-direction returns to zero, the phase measurement also returns to its initial value.

This relationship between the g-force measurements in the y-direction and the corresponding phase measurements further emphasizes the interconnected nature of the microbridge response and the finger movement.

#### 3.3.2. Finger Movement: Magnitude Shift Detection

To measure the magnitude shift, we utilized the maximum point on the magnitude curve as the fixed frequency measurement corresponding to the device resonant frequency. When mass is added to the bridge, a decrease in the magnitude at this specific frequency point can be observed, as depicted in Figure 8.

The slope of the magnitude curve after the resonance point is approximately −1.2 × 10^−4^ dB/Hz. As the finger is moved closer to the sensor by a few millimeters, the magnitude decreases by approximately 0.15 dB. This change is equivalent to a frequency shift of approximately 1.25 kHz. In terms of proportionality to the resonance frequency, this shift amounts to around 100 ppm, indicating a slightly lower sensitivity compared to the phase detection method, as expected: at the resonant frequency point, the slope of the phase curve is maximum; however, this is not the case for the magnitude curve, as it is already close to its maximum point and, therefore, exhibits a relatively small slope.

Similarly, when the finger is moved further away, the magnitude recovers rapidly, as does the phase detection method. The slope of the magnitude recovery is approximately 0.8 dB/s, corresponding to a frequency shift of approximately 6.6 kHz per second.

### 3.4. Breath Detection

The last experiment that we performed is breathing 10 cm from the sensor with a regular (Figure 9) and irregular (Figure 10) respiration rate. For this experiment we used phase detection, fixing the measurement of the phase at the resonant frequency point. We can see in Figure 9 that, when we exhale air over the sensor, the phase decreases abruptly due to the shift to low frequencies, which is a result of the increase in humidity. After each exhalation, the phase recovery is not complete. After the last exhalation, the phase takes roughly four seconds to abruptly recover the initial value (the last jump).

This abrupt recovery of the initial value is now under discussion, but it seems that we can have a water layer on the bridge which evaporates progressively until a system of water islands is formed. The water islands increase the evaporation velocity and, together with the fact that the mass remaining on the bridge in this situation is too small, causes the final phase jump. Figure 10 shows an irregular breath pattern, demonstrating the ability of the sensor to clearly follow the breath rates. 

## 4. Conclusions

In this study, we have introduced a simple sub-micrometric metal bridge integrated with CMOS technology as a promising candidate for a fast and highly sensitive non-contact human activity sensor. The sensor leverages its ability to detect humidity resulting from water evaporation on the body’s surface. Both phase and magnitude changes can be monitored for this purpose.

In the case of phase change, moving the finger closer or further away from the sensor induces a frequency shift of approximately 14 kHz (as shown in Figure 5). This frequency shift corresponds to a relative humidity modification of around 4%, indicating a high sensitivity of the sensor. Taking into account the frequency of resonance and the calibration performed in [15], our sensor presents a sensitivity of 0.03% of variation of frequency per %RH, an average response time of 1 s, and 2 s of average recovery time.

Although we conducted the experiments using a network analyzer in an open-loop mode, it is easily adaptable to a closed-loop mode using a frequency counter. Despite the small quantity of water adsorbed by the metal, the high resonant frequency, good mass sensitivity, and readout circuitry enable excellent human activity detection. Additionally, the limited amount of water adsorbed facilitates a quick recovery of the phase or magnitude to their previous values, enabling rapid movement detection. Good correspondence between our sensor for finger movement detection with a commercial gyroscope has been demonstrated with the advantage of providing a non-contact measurement system.

Other authors referenced here have presented sensors with high sensitivity. As far as we know, sensors based on surface acoustic wave resonators are the best. For example, the authors of [10] have obtained a sensitivity of 0.005% of variation of frequency of resonance per %RH. The authors of [17] have proposed a 4-anchored plate oscillator with a variation of 0.02% of frequency of resonance per %RH. These two examples have a better sensitivity than our sensor, but a complex fabrication is needed, specifically for SAW sensors. Further, technical uncertainties can appear when using the proposal outlined in [17], due to the use of distributed holes in the plate to accomplish resonator releasing.

In [18], the authors have obtained a sensor comparable with our proposal in response and recovery times. For the response time, the authors of [18] have obtained 1 s as in our case. For recovery time, the authors have obtained 2.8 s, which is very close to the 2 s of our proposal. However, in [18], a film of MoS_2_ is needed, whereas in our proposal no absorbing layer is needed.

In comparison with [17,18,19,20], our metal bridge MEMS resonator has a good sensitivity in the minimum detecting area, even without the use of a specific sensing layer over the resonator.

Overall, this integrated metal bridge sensor demonstrates the potential to serve as a highly sensitive and fast human activity sensor, making it a valuable tool for various applications.

## Figures and Tables

**Figure 1 sensors-23-08945-f001:**
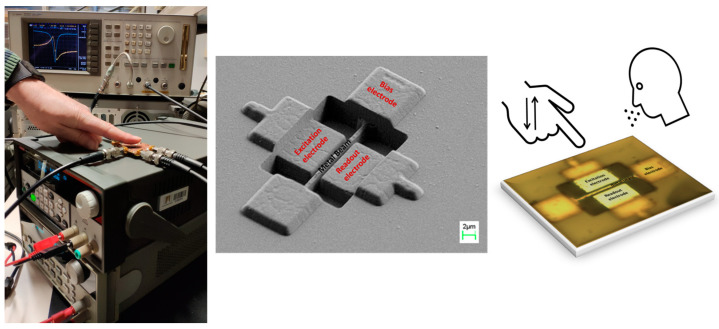
(**Left**) Picture of the set-up used to measure the signal coming from the resonator (clamped-clamped beam) and the movement of the fingertip. (**Center**) SEM image of the metal bridge resonator before releasing process with the different relevant parts highlighted (sample as received from the foundry). All the electrical connections to the CMOS circuitry are underneath the CMOS protection layer surrounding the resonator area. (**Right**) Optical image of the released bridge with sketch of the humidity environment change due to breath or finger proximity.

**Figure 2 sensors-23-08945-f002:**
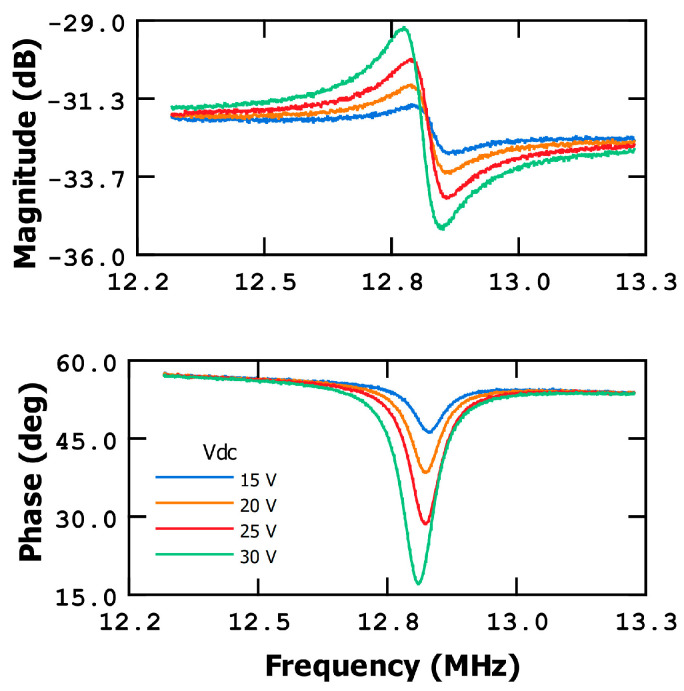
Frequency response of metal bridge at different Vdc levels in open-loop and two-port configuration.

**Figure 3 sensors-23-08945-f003:**
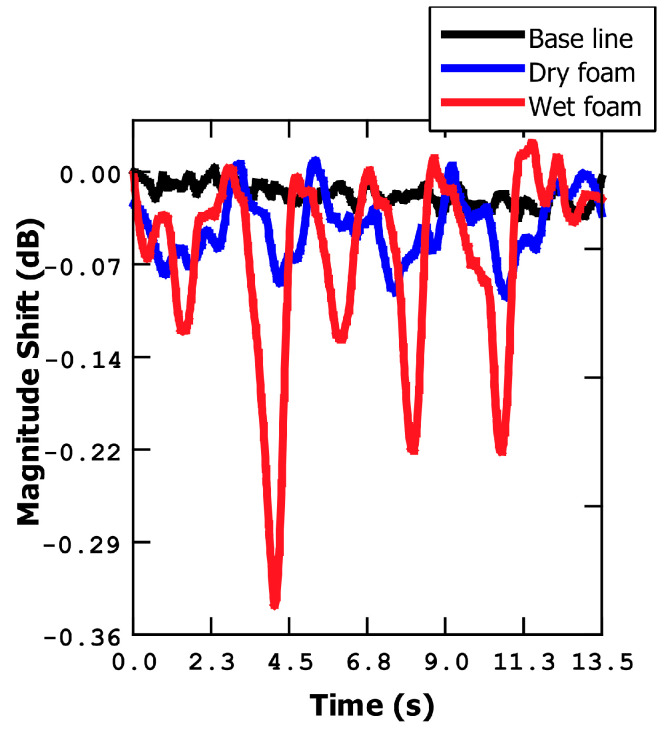
Magnitude shift at a fixed frequency (resonance frequency) bringing foam closer and further away from the metal bridge. Results for dry and wet foam.

**Figure 4 sensors-23-08945-f004:**
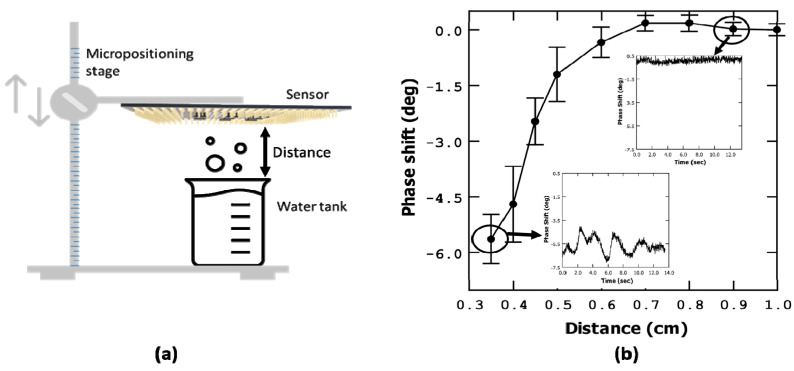
Phase shift versus chip-to-water tank distance, measured at the resonant frequency. (**a**) Set-up used to perform the measurement. (**b**) Phase shift versus distance plot. In the insets of (**b**) oscillations over time of the phase measurement are shown.

**Figure 5 sensors-23-08945-f005:**
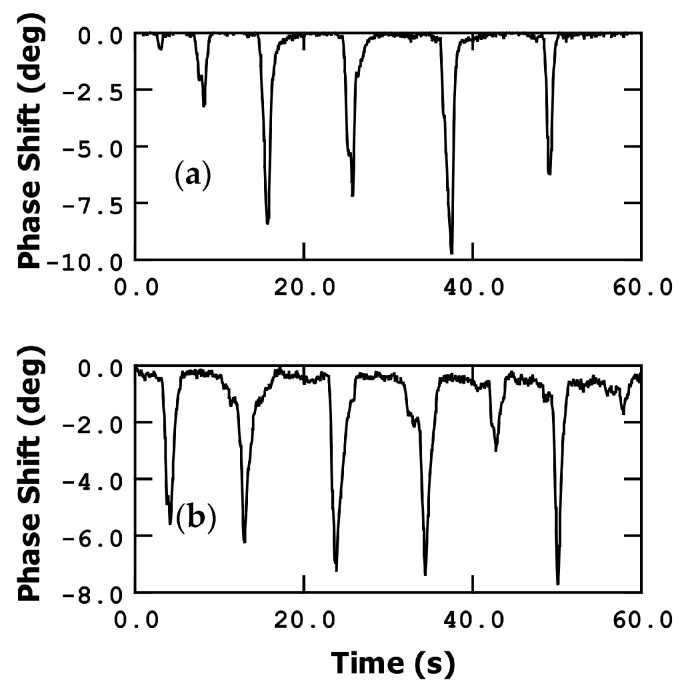
Finger movement detection measuring phase shift. (**a**) Up and down movement; (**b**) left and right movement.

**Figure 6 sensors-23-08945-f006:**
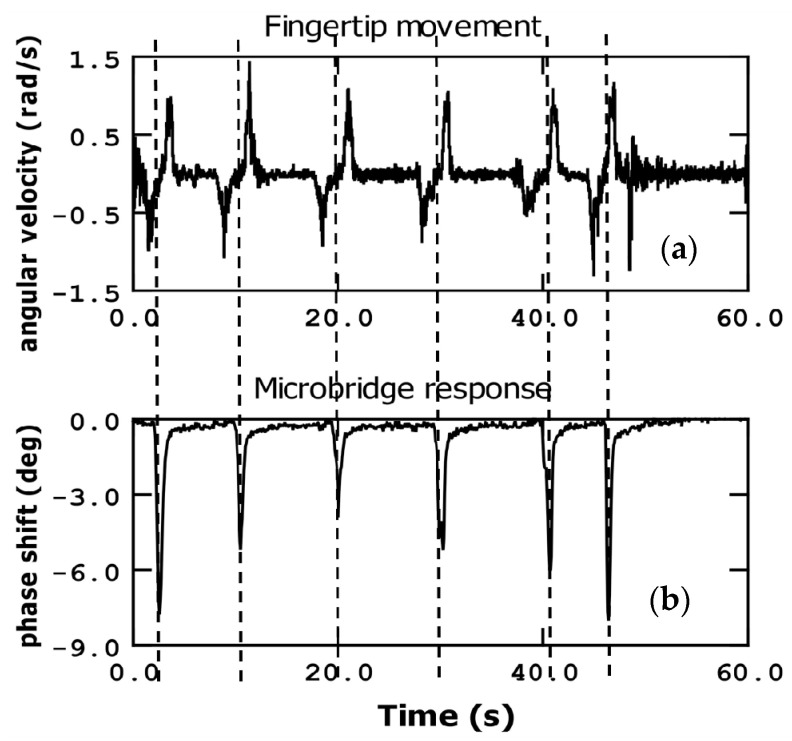
Fingertip movement (up and down) sensed by a commercial gyroscope measuring the angular velocity (**a**) in comparison with the detection by the microbridge sensor (phase shift) (**b**).

**Figure 7 sensors-23-08945-f007:**
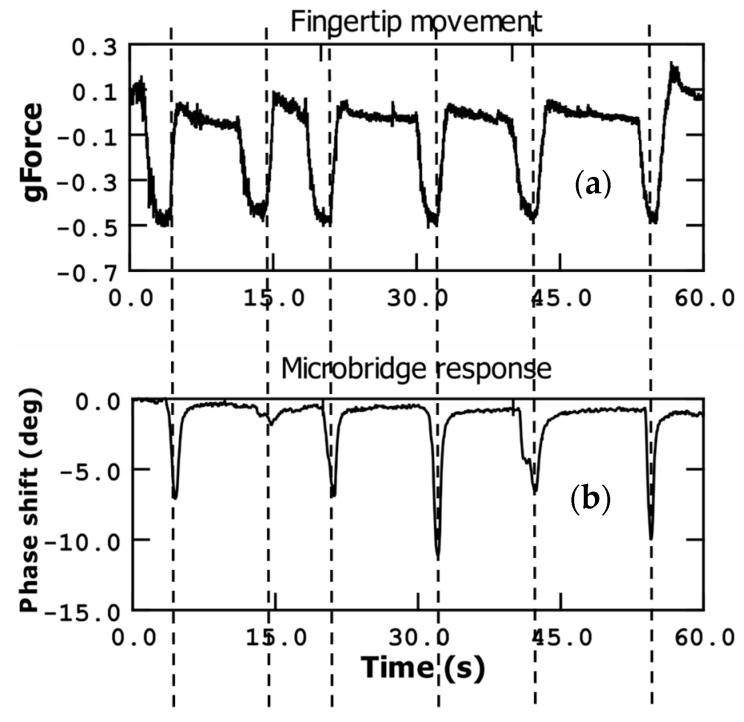
Fingertip movement (up and down) sensed by a commercial accelerometer measuring g-force (**a**) (axis perpendicular to earth gravity direction) in comparison with the detection by the microbridge sensor (phase shift) (**b**).

**Figure 8 sensors-23-08945-f008:**
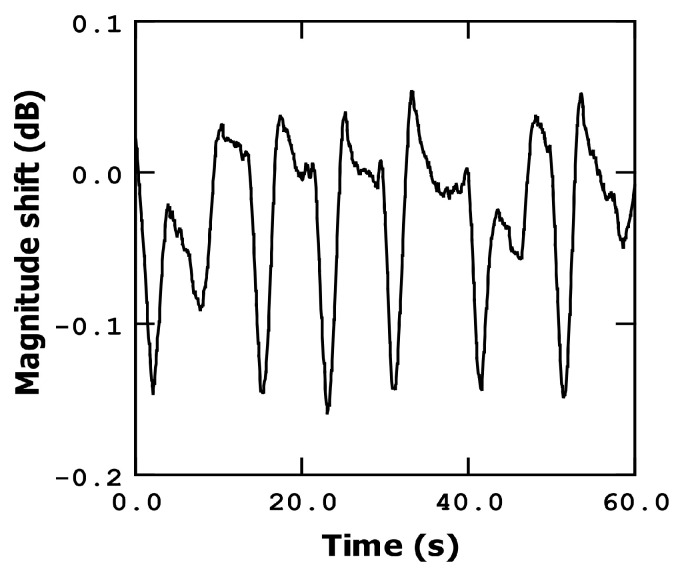
Fingertip movement (up and down) detection using magnitude shift.

**Figure 9 sensors-23-08945-f009:**
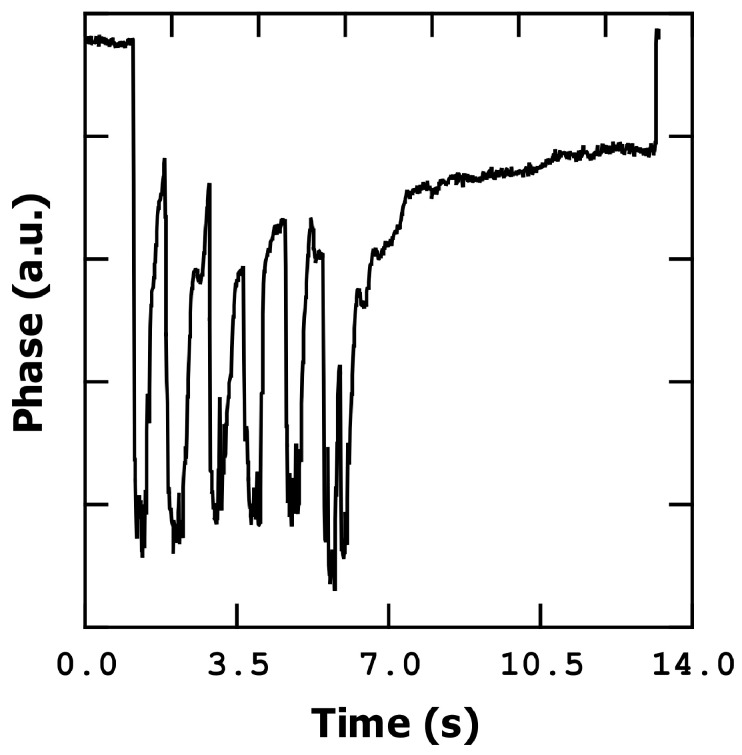
Phase change during six seconds of regular breathing.

**Figure 10 sensors-23-08945-f010:**
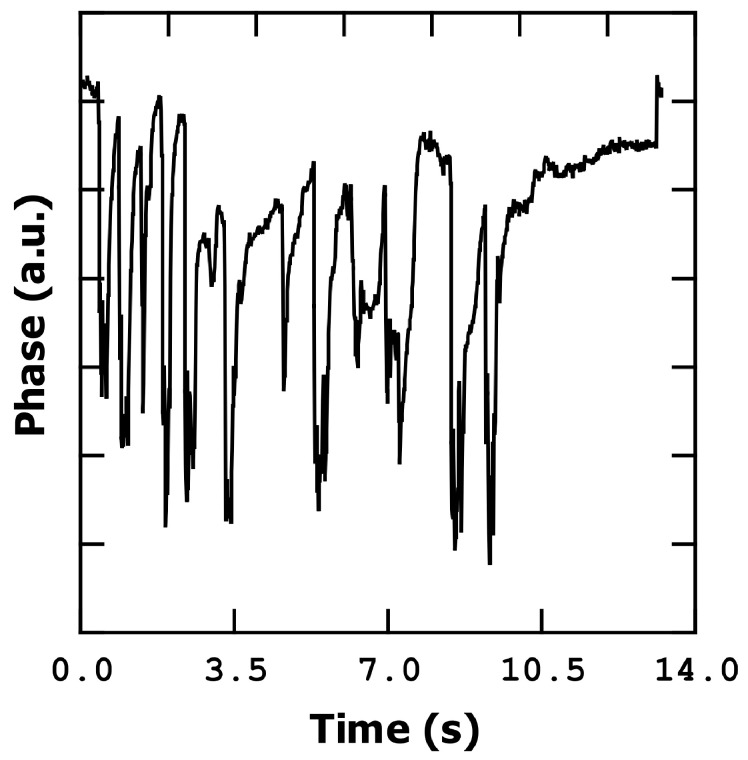
Phase change during nine seconds of irregular breathing.

## Data Availability

Data supporting figures within this paper are available from the corresponding author upon reasonable request.

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
