# Peer review of "Metal Microelectromechanical Resonator Exhibiting Fast Human Activity Detection"

_sensors, 2023, doi:10.3390/s23218945_

Round 1

Reviewer 1 Report

Comments and Suggestions for Authors

Dear Editor:

Thank you for giving me this opportunity to review the manuscript. English language is fine and Experiment is well written in this manuscript.

Questions

1.     It is necessary to express the content of this article in abstract. Please improve it.

2.     Please rewrite introduction part. It should include investigation of the research content of this  manuscript.

3.     Measurement theory is required.

4.     Experiment is enough, but it is short of discussion part and test analysis.

Author Response

Please see the attachment, thanks in advance!

Reviewer 2 Report

Comments and Suggestions for Authors

Author Response

(The authors gave the same response as above.)

Reviewer 3 Report

Comments and Suggestions for Authors

This paper presents a simple sub-micrometric metal bridge integrated with CMOS technology, which is a promising candidate for a fast and highly sensitive non-contact human activity sensor. Several concerns are provided below.

1. The introduction part is too brief and should provide a literature review of the current level of development of some of the relevant technological processes. Also, the human activity sensor should also be researched in detail.

2. The sensor design section is too briefly written and could provide a bit of theory and more technical approaches.

3. Some figures are not clear, which should be impoved, for example, Figure 4.

4. The authors may compare the performance of various aspects of the sensor in this work with similar published designs to demonstrate the innovation and advantages of this design.

Author Response

(The authors gave the same response as above.)

Reviewer 4 Report

Comments and Suggestions for Authors

This is an interesting approach application for a MEMS resonator. 

Sections 3:  Could the authors add details of specific equipment used for the tests?

Can the authors comment more on what physical properties are changed by humidity? I assume that the humidity affects the thermal conduction path?

Where the authors able to perform calibration of the senor in a humidity chamber, or using a separate sensor?

Comments on the Quality of English Language

The style of English is good. Just a minor check required. 

Author Response

(The authors gave the same response as above.)

Round 2

Reviewer 1 Report

Comments and Suggestions for Authors

Dear Editor:

Thank you fou your email.

Question: Figures includin measurement device and test method are necessary(seen as follows).

Author Response

Dear Sir,

Following your suggestions, we have modified Figure 1 in order to include in it a picture of our measurement set-up.

Reviewer 3 Report

Comments and Suggestions for Authors

Thanks to the authors for the revisions. My concerns have been addressed.

Author Response

Thank you for your suggestions.